# Web Site Fingerprint Attack Generation Technology Combined with Genetic Algorithm

**Hanfeng Bai [1,*], Junkai Yi [1,*] and Ruidong Chen [2]**

1 Key Laboratory of Modern Measurement and Control Technology Ministry of Education, School of Automation, Beijing Information Science & Technology University, Beijing 100096, China
2 Center for Cyber Security, University of Electronic Science and Technology of China, Chengdu 610054, China
* Correspondence: baihf0511@foxmail.com (H.B.); yijk@bistu.edu.cn (J.Y.)

**Abstract:** An anonymous network can be used to protect privacy and conceal the identities of both communication parties. A website fingerprinting attack identifies the target website for the data access by matching the pattern of the monitored data traffic, rendering the anonymous network ineffective. To defend against fingerprint attacks on anonymous networks, we propose a novel adversarial sample generation method based on genetic algorithms. We can generate effective adversarial samples with minimal cost by constructing an appropriate fitness function to select samples, allowing us to defend against several mainstream attack methods. The technique reduces the accuracy of a cutting-edge attack hardened with adversarial training from 90% to 20–30%. It also outperforms other defense methods of the same type in terms of information leakage rate.

**Keywords:** anonymous network; fingerprint attack; genetic algorithm; counter sample

## 1. Introduction

In today's world, with the rapid development of the Internet, the efficiency of people's access to information has been greatly improved, and the protection of online privacy has become more and more important. Universities, scientific research institutes, government agencies and privacy-focused online activists all use anonymous software to carry out online activities in order to protect the potential risk of privacy disclosure. Tor, as a classic representative, is the most popular anonymous software today. It provides network anonymity technology for users by, primarily, avoiding the leakage of users' IP information and browsing behavior. Figure 1 shows a schematic diagram of the Tor network architecture.

Tor's anonymous network builds anonymous access communication links through a group of three relay nodes, which can effectively avoid the privacy leakage risk caused by the attack of forwarding nodes in the anonymous service of a single proxy mode. However, due to different types of web content, such as text web, video web, file transfer, etc., the data distribution is not the same. The site fingerprint of the anonymous network is used to extract the distribution characteristics of traffic data when users visit web pages and use them for the identification of specific types of websites.

When an anonymous web user browses the Web, the browser will download a variety of web resource files. At the same time, the size of web resource files is different among different types of websites. Taking Microsoft Edge as an example, each file transfer and load requires a separate TCP connection; therefore, it is easy for an attacker to obtain the data stream loaded by a single file and analyze the characteristic data. When the user browsed the web, the attacker calculated the file size transmitted by the encrypted proxy website to the user's computer port, and processed the timely data via proxy encryption and other means, and still could obtain the size and number of files received by the user, which was the website fingerprint based on traffic analysis in the general sense.

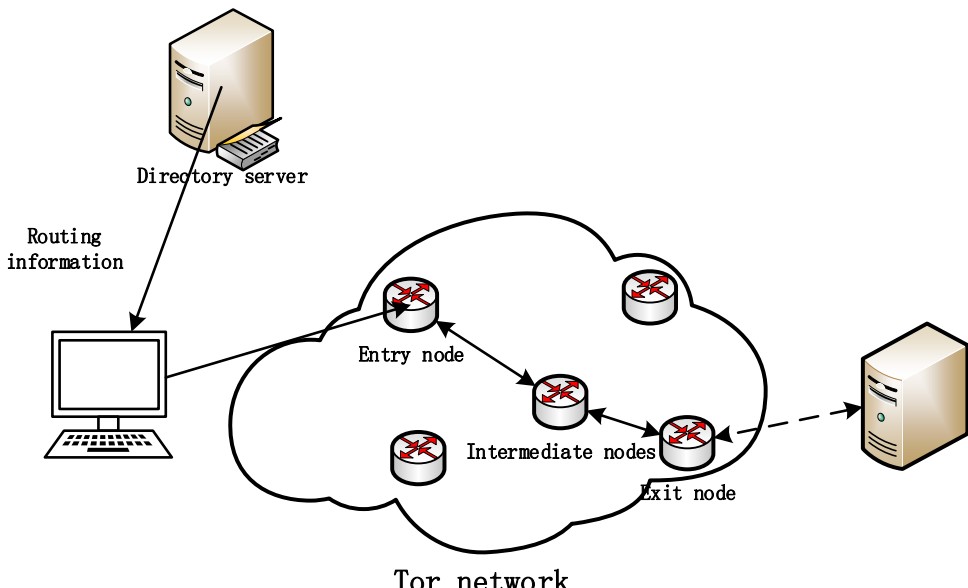

**Figure 1.** Schematic diagram of Tor network architecture.

Tor has created many encryption methods after years of development. When the data collected by an attacker are encrypted, the timestamp, size and direction information of the packet will still be exposed. Using the patterns in this information, often with the help of machine learning, an adversary can often uncover the client's activity.

However, due to the characteristics of Tor, it is difficult to confirm communication relationships and monitor user behavior. Traditional traffic analysis methods based on traffic tracing and packet content analysis gradually become ineffective; thus, it is difficult to obtain evidence for crimes committed using Tor. Website Fingerprinting (WF) attacks use the order, direction and size of the packages used to load the pages so that a website can be identified as unique, enabling traffic eavesdropping or fingerprinting attacks to detect where traffic is coming from, render anonymous networks useless and cause the disclosure of private information. Strong defense systems preventing this, however, can be costly. Recent WF research has focused on creating padding schemes that remain effective when the bandwidth and latency overhead remains low. Unfortunately, these lightweight defenses may fail when new attacks [1] are introduced; implementation challenges [2] may also make deployment impractical. In order to combat these attacks, the researchers proposed to modify the anonymous traffic and implement fingerprint obfuscation to reduce the classification accuracy.

The methods for defending against fingerprinting attacks on anonymous websites have many problems including limited defense range, poor mobility and excessive performance overhead. This paper proposes a traffic camouflage technology, which uses an adversarial generative network based on Wasserstein distance combined with a genetic algorithm. The main contributions of this paper are as follows:

- We propose WGAN-GA, the WF defense, to leverage the concept of adversarial examples. Our evaluation shows that WGAN-GA can significantly reduce the accuracy of WF attacks from 98% to 20–50% and that it performs better on a variety of models.
- In terms of algorithm efficiency, the extra bandwidth usage is 1% for our method and 0.8% for the DE (Differential Evolution) algorithm. Our convergence time, however, is shorter.
- Using the WeFDE framework [3], we measure the information leakage of WGAN-GA, and find that it has less leakage for many types of features than either Walkie-Talkie (W-T) [4] or WTF-PAD [5].

## 2. Threat and Defense Model

### 2.1. Attack Model

We assume that clients browse the Internet through the Tor network to hide their activities. Figure 2 is a schematic diagram of the fingerprinting attack model. The critical adversary is local, meaning that the attacker is located somewhere in the network between the client and the Tor guard node. It is assumed that the attacker already knows the identity of the client. His goal is to detect which website the client is visiting. The local adversary can be an eavesdropper on the user's local network, a local system administrator, an Internet service provider, or an operator of any network or ingest node between the user and the ingest node. The attacker is passive and only observes and records the traffic traces flowing through the network. He has no ability to drop, delay or modify real packets in traffic.

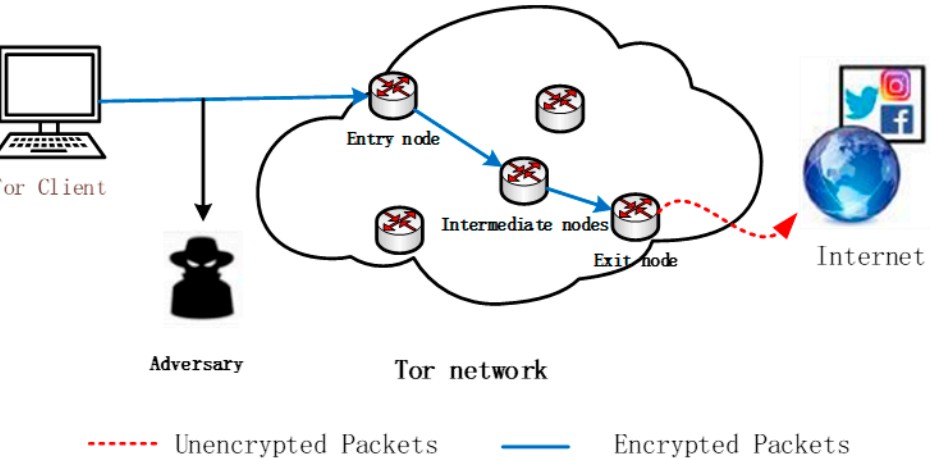

**Figure 2.** Website fingerprinting attack model.

In a website fingerprinting (WF) attack, the attacker feeds collected web traffic into a trained machine learning or deep learning classifier. For training, the WF attacker first needs to collect the traffic of individual sites by manipulating the Tor client. Since it is not possible to collect traffic from all websites, the attacker identifies a set of monitored websites that he wants to track. The attacker limits the attack to identifying any website access within the monitored set. The set of all other sites is called the unmonitored set.

WF attacks and defenses are evaluated in two different environments: closed-world and open-world. In the closed-world setting, we assume that customers are limited to accessing monitoring sites, and the training and test sets used by the attacker contain only samples from the monitoring set. The closed-world scenario models an ideal environment for the attacker and does not reflect the attacker's performance in the real world. Thus, from the perspective of developing WF defenses, demonstrating its ability to defend against closed-world attacks is sufficient to show its effectiveness.

### 2.2. Defense Model

The purpose of WF defense is to prevent an attacker from observing a traffic trace, and, thus, prevent the attacker from accurately determining to which site the trace belongs. In order to achieve this, the real traffic has to be manipulated somehow. Since traffic is bidirectional, the deployment of a successful WF defense requires the participation of both the client and a cooperating node in the Tor circuit. We refer to this node as the bridge. To prevent an eavesdropper from performing a WF attack, the bridge can be any node between the attacker and the client's target server, so that the attacker can only access obfuscated traffic. Since the protection node knows the IP address of the client and can act as a WF adversary, it is better to establish a bridge at the intermediate node, stopping the adversary from identifying the client directly.

## 3. Related Work

Traffic analysis can carry out side channel attacks in a variety of scenarios [6–8]. Fingerprint attacks based on network traffic analysis are a common attack method and include system fingerprints and application fingerprints. Recent research work has shown that advanced anonymous mechanisms are still vulnerable to these attacks; thus, this paper proposes a new defense mechanism.

### 3.1. Fingerprint Attack and Defense Technology for Anonymous Network

A fingerprint attack is usually modeled as a classification problem. The attacker extracts features from the side channel and classifies a group of websites/applications [9,10]. Dyer et al. [11]. first conducted a comprehensive study on traffic analysis attack and defense in 2012, and their results showed that many countermeasures at that time were unable to resist even simple attacks using general traffic characteristics (such as bandwidth and total time). Wang et al. [12] constructed a k-nearest neighbor (KNN) classifier with a large number of feature sets. Their classifier has been tested in a large open-world environment and performs better than defenses in some previous studies. The literature [13–15] proposes that lateral channel data can effectively reveal enough information. Hayes et al. [16] proposed an attack using a random forest model to generate website fingerprints, and launched a website fingerprint attack using a large amount of noise data in the Tor environment. Panchenko [17] used an SVM (Support Vector Machine) classifier to classify CUMUL (Cumulative Sum Fingerprinting) features of different streams. At the same time, they used representative data to assess fingerprints on an Internet scale. However, their experiments showed that the existing methods could not be applied to any web page in their data set. In recent studies, methods using more advanced models, such as deep learning models (supervised model [18] and unsupervised model [19]), have been investigated, and fingerprint printability has been measured and analyzed. Liu et al. [20]. first applied PHMM (Profile Hidden Markov Model) to website fingerprint attacks. In this way, an attacker can model a single web page or combine different web pages from the same site to the model. The literature [21] also proposes a website fingerprint attack method, which uses a deep convolutional neural network, with better accuracy and recall rates than the previous deep learning algorithm.

The core idea of the adversarial sample generation algorithm is to add difficult-to-distinguish interference to the sample, so that the classification model can judge the wrong result. For traffic, the adversarial sample generation algorithm attempts to make the traffic detection model appear as the traffic of other categories.

Adversarial samples are special samples designed by the attacker to input into the deep learning model and cause the model a classification error. This kind of adversarial sample designed for deep learning models has attracted the attention of many researchers in recent years. Walkie-Talkie [4] fills fake Tor units for website fingerprints, which consumes some extra network traffic. DE is a kind of defense method against fingerprint attacks proposed in the literature [21]. Differential Evolution (DE) is used to fill and modify the website fingerprint, which achieves remarkable defensive effect. Consider that the goal of the counter sample is to misjudge the detection model. This meets the purpose of traditional traffic obfuscation to some extent; that is, the adversarial traffic is generated based on the actual traffic, and then the adversarial traffic is forwarded to achieve traffic confusion.

### 3.2. Genetic Algorithm

A genetic algorithm can find the feasible solution space of the problem, and then find the possible optimal solution, which is the uncertainty optimization in optimization problems. Uncertain optimization relies on random variables in the direction of search, rather than on a mathematical expression. Compared to other algorithms, when the optimization converges to the local extreme value, the search results can jump out of the local optimal solution and it can continue to search for a better feasible solution.

By selecting the appropriate objective function, the problem of generating adversarial samples can be transformed into an optimization problem and solved. The process of solving the optimal solution corresponding to the objective function is actually the process of generating antagonistic samples. This shows that a genetic algorithm can be effectively applied to parameter optimization and function solving in machine learning and other fields. In terms of parameter optimization, Chen et al. used a parallel genetic algorithm to optimize parameter selection of a support vector machine (SVM) [10]. Experiments show that this method is superior to a grid search in classification accuracy, feature selection quantity and running time. The method uses a genetic algorithm to select the feature set with the highest detection rate. By selecting the appropriate objective function, the generation of adversarial samples is transformed into the solution of an optimization problem.

The previous defense method based on an adversarial generative network has some shortcomings with the development of attack technology, such as poor robustness, weak transferability and lack of black box. After reviewing the relevant literature, it is found that the GA algorithm has strong robustness, which is attributed to the operation of repeatedly crossing and re-evaluating individuals in the algorithm. It has also been used in some studies. On this basis, we conducted experiments and found that GA has improved on such problems.

## 4. Method

In order to resist the fingerprint attack of website traffic, researchers have carried out ample research, but most of the existing schemes are white box methods, suffer from a lack of dynamic and migration, and the same sample may generate the same result, which is a big hidden trouble for the defense side. Moreover, most schemes are difficult to play a role in once monitored. Therefore, a black box attack method that can be dynamically disguised is needed. This section mainly illustrates the construction of an adversarial network and a genetic algorithm.

### 4.1. Overall Structure

The principle of a fingerprint attack algorithm is to carry out pattern matching on captured data packets and match the traffic data to a certain website. Through the above identification matching method, the anonymity is destroyed. In order to bypass the detection of a fingerprint attack model, an adversarial sample generation method based on traffic data is proposed, and the generated perturbed genetic algorithm is screened. This method uses the concept of counter sample in deep learning to obfuscate traffic, in order to bypass fingerprint attack model detection. The overall structure of the system is shown in Figure 3.

### 4.2. WGAN-GA Traffic Generation Model

In order to increase the dynamics and mobility of the existing fingerprint attack methods of anonymous websites, in this paper, WGAN with gradient punishment combined with genetic algorithm is used to generate counter samples to defend against fingerprint attack models.

#### 4.2.1. Basic Principles of Adversarial Networks

GAN is a result formed after two models play against each other and meet certain requirements. The generator *G* learns data characteristics from the training data, and the discriminator judges whether the data are generated by *D* or the original training data. *G* and *D* play a confrontational game. At the end of this game, there is only one optimal solution: the generator *G* learns the distribution of training data and generates samples that are indistinguishable from these training data. The discriminator cannot classify the generated samples [22].

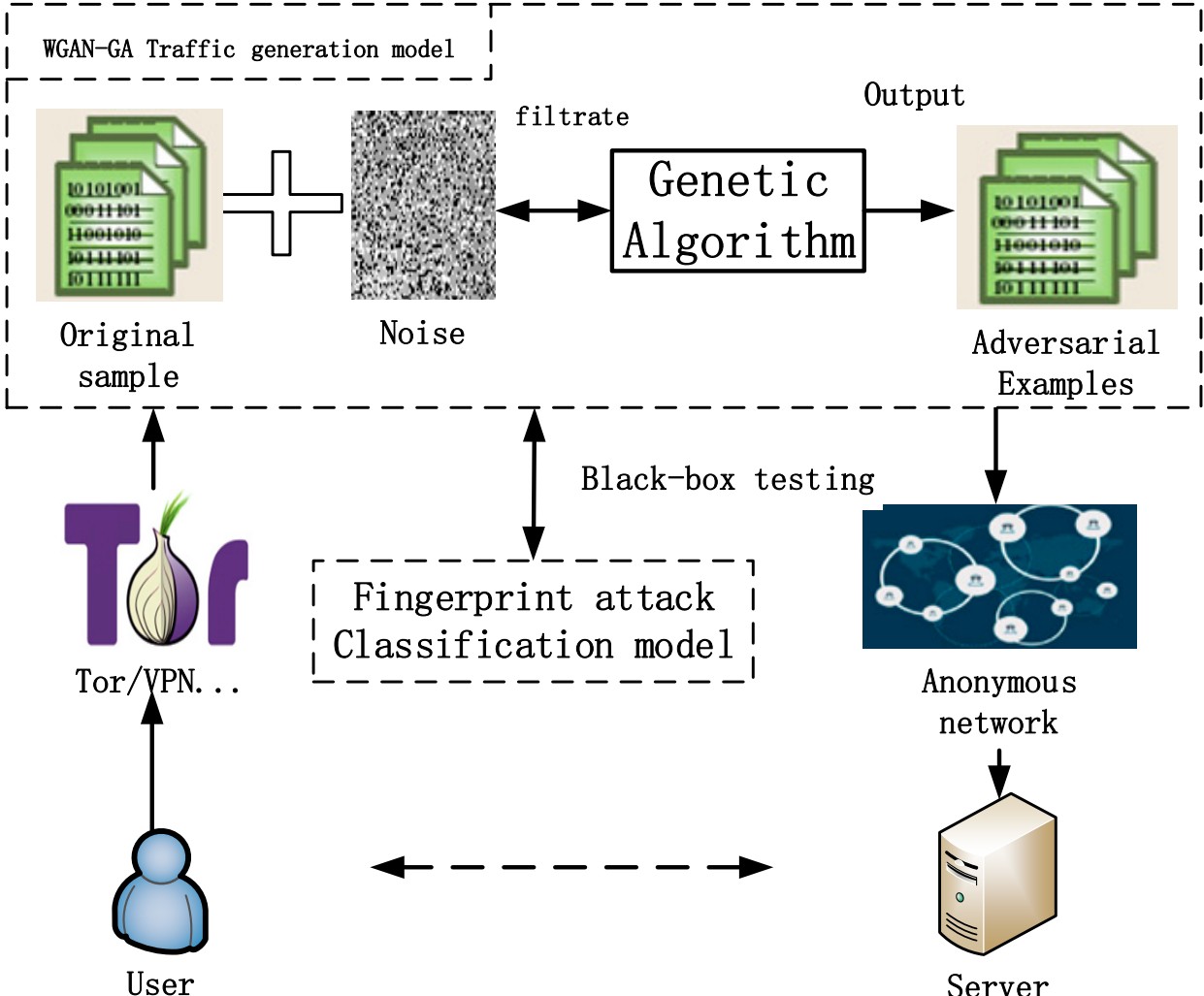

**Figure 3.** Overall system structure diagram.

The loss function *J(G,D)* is used for min-max model for GAN.

$$\min_{G} \max_{D}(J, D) = E_{x \sim p(data)}[Log_a D(x)] + E_{z \sim P_g(z)}[1 - Log_a D(G(z))] \tag{1}$$

The classical GAN network model uses JS (Jensen–Shannon) divergence optimization training parameters, and the input data of its generator is Gaussian noise. The loss function is shown in Equation (2):

$$J_G = -E_{z \sim P_g(z)}[D(G(z))] \tag{2}$$

Input data of discriminator are generated data and real data of generator, and its loss function is shown in Equation (3):

$$J_D = -E_{x \sim p(data)}[Log_a D(x)] + E_{z \sim P_g(z)}[1 - Log_a D(G(z))] \tag{3}$$

where $J_G$ represents generator loss function; $J_D$ represents discriminator loss function; $E[.]$ represents the expected function; $G[.]$ represents the generator function; Base Log is usually 2, 10 or the natural constant e; $D[.]$ represents the generator function. *p(data)* represents the data distribution of the sample; is noise data distribution; z is the input noise data vector.

GAN model is prone to gradient disappearance during training, which leads to model collapse. To solve this problem, some researchers use Wasserstein distance instead of JS (Jensen–Shannon) divergence as a cost function. However, gradient explosion and non-convergence often occur in WGAN in practical applications. Therefore, the penalty term

is added to the loss function to satisfy the Lipschitz continuity condition. To make up for the defects of WGAN Network, the improved model is called WGAN-GP (Wasserstein Generative Adversarial Network with Gradient Penalty). The loss function $L(G,D)$ of WGAN-GP model can be expressed as Equation (4):

$$\min_{G} \max_{D}(J, D) = E_{x \sim p(data)}[D(x)] - E_{z \sim P_g(z)}[D(G(z))] + \lambda E_{\hat{x} \sim P_{\hat{x}}(\hat{x})}[\|\nabla \hat{x} D(\hat{x})\|_p - 1]^2 \tag{4}$$

where represents p norm; represents gradient operator; $\lambda$ penalty is the coefficient; $\lambda = \varsigma \, x + (1 - G(z))$, $\varsigma$ obey uniform distribution within the scope of [0, 1]; is the sample data distribution $P(data)$ and linear uniform sampling between sampling points in the generated data distribution.

In the training process of WGAN-GP model, the generator inputs Gaussian noise of the same dimension as the measured information and takes the normal measured information as the target data. The generator and discriminator conduct game confrontation training based on Equation (4).

### 4.2.2. Perturbation Screening Method

As long as the statistical characteristics of the original data remain unchanged, interference is added to the network traffic data; in this way, the traffic detection model will misjudge it as the traffic of other categories. The type and amplitude of disturbance features are limited to ensure the "validity" of the samples. The genetic algorithm is used to screen the disturbance degree of the features, in order to realize the black-box countermeasure sample generation method for the traffic fingerprint attack model.

For example, suppose the original traffic is:

$$X_i = [X_i^0, X_i^1, \cdots, X_i^n] \tag{5}$$

After coding, it is:

$$P_i = [P_i^0, P_i^1, \cdots, P_i^n] \tag{6}$$

The range of each dimension of the original flow is:

$$S_j = [lower_j, upper_j] \; j \in [0, n] \tag{7}$$

When =0, it indicates that the mutation probability of the *i*th feature of the original sample is 0. When it is positive, the sample changes in the positive direction. In sum, the calculation formula of samples after each iteration is shown in Equation (8):

$$X_i^j = \begin{cases} X_i^j + P_i^j(X_i^j - lower_j) & P_i^j > 0 \\ X_i^j + P_i^j(upper_j - X_i^j) & P_i^j \leq 0 \end{cases} \tag{8}$$

Finally, the fitness function formula of the genetic algorithm is obtained as shown in Equation (9):

$$F(P) = c_1 D(P) + c_2(1 - E(P)) \tag{9}$$

$$D(P_i) = \sqrt{sum_j^n \left( X_i^j - X_i^{j'} \right)^2} \; j \in [0, n] \tag{10}$$

$D(P)$ is the distance between the original sample and the opposing sample. In this paper, L2 norm is used, and $E(P)$ represents the attack success rate of the current sample.

## 5. Experiments

In order to verify the defense effect of the proposed method against website fingerprint attack methods, more advanced WFP models in recent years were used for evaluation,

namely, PHMM [9], CUMUL [11], K-FP [16] and DF [1], and experiments were conducted on data sets.

### 5.1. Data Set

Since this paper studies the fingerprint attack defense technology for anonymous websites, the preprocessed fingerprint data set of anonymous websites by Sirinam et al. [1]. is adopted, which consists of 100 monitored websites and contains 1000 instances in total. The data set is a public data set collected for anonymous networks, and users can only access the monitored website set. Each instance is a packet direction sequence, represented by a 5000-dimensional vector. It is defined as Packet Byte Vector (PBV). The data set was randomly divided into a training set, a verification set and a test set according to the ratio of 8:1:1.

### 5.2. Model Construction and Training

#### 5.2.1. Model Structure

In this paper, the generator and discriminator of the adversarial network model are built by a multi-layer perceptron, a fully connected network, and Keras as the back end of TensorFlow. Figure 4 shows the detailed structure of the generator and discriminator network.

The parameters of the genetic algorithm are shown in Table 1:

**Table 1.** Parameters of genetic algorithm in this paper.

| Parameter | Value |
|---|---|
| Selection operator | Tournament method |
| Crossover probability | 0.5 |
| Probability of variation | 0.3 |
| c1 | 1000 |
| c2 | 150 |

#### 5.2.2. The Influence of Population Size on Genetic Algorithm

Population size is an important parameter of a genetic algorithm, which plays a crucial role in the algorithm operation results. If the population size is too small, it may fall into the local optimal solution. Otherwise, the algorithm convergence speed may be too slow. In this experiment, the system energy is greatly reduced so that the system availability is reduced.

According to the above network parameters, the antagonistic samples of the department are produced, and then the genetic algorithm is used for screening. The data are pre-tested in Table 2, and the algorithm running time and the fitness values of different population sizes are given. As can be seen from the overall change trend, the fitness of the sample gradually increases with the increase in the population, and the algorithm running time also increases proportionally. It is noted that when the population size increases from 150 to 200, the running time increases by about 50%, while the fitness value only increases by 8%.

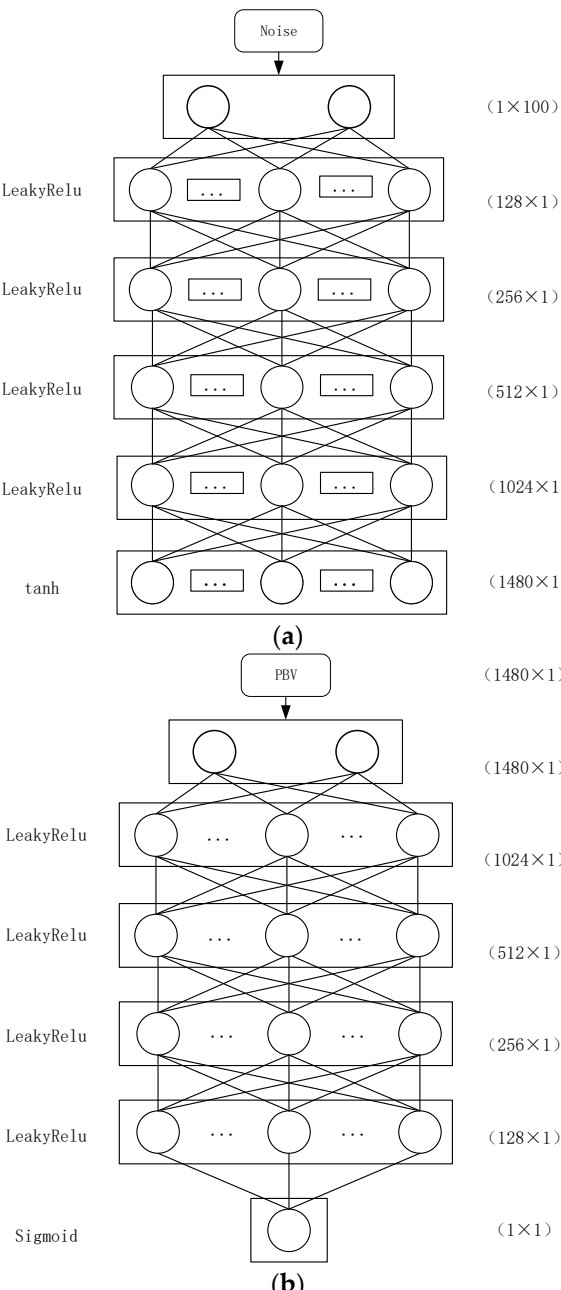

**Figure 4.** Network structure. (**a**) Generator network structure; (**b**) discriminator network structure.

**Table 2.** The effect of population on algorithm parameters.

| Population Number | Algorithm Running Time | F(P) |
|---|---|---|
| 50 | 29.5 | 7894.6 |
| 100 | 37.2 | 10,567.1 |
| 150 | 53.4 | 12,435.7 |
| 200 | 76.5 | 13,855.5 |

Based on the analysis of the above experimental results, it can be concluded that the selection of the appropriate population number is crucial to the algorithm result. Considering the performance of sample generation and the running time of the algorithm, the selection of the population number is 150.

### 5.2.3. Training Process

During training, train generators and discriminators alternately. For example, when the discriminator is trained, the generator parameters are fixed, the loss function of the generator is obtained, and the discriminator parameters are updated by backward propagation. To maintain counterbalance, set the discriminator and generator iterations to 2:1. At the same time, the L2 norm is used to represent the similarity between the admissible sample and the original data. After the criterion and the similarity measure of disturbance success are determined, the genetic algorithm is used for screening.

The loss function of the discriminator adopts gradient penalty Wasserstein distance and the loss function of the generator adopts Wasserstein distance.

### *5.3. Effect Evaluation*

Since it is difficult to implement the defense strategy on an anonymous network client, the general evaluation method of fingerprint attack defense research is adopted to attack the data set processed by the defense method, and the classification accuracy is obtained.

### 5.3.1. Evaluation Metrics

The accuracy of common evaluation indicators of deep learning models is used as the evaluation index, and the relevant formula is shown in Equation (11).

$$Acc = \frac{TP}{TP + FN} \tag{11}$$

where *Acc* represents the classification accuracy of the model; *TP* represents the number of positive samples correctly classified; *TN* represents the number of negative samples correctly classified; *FN* represents the traffic that belongs to the positive sample category and is classified into the negative sample category.

### 5.3.2. Defensive Effect

Defense tests were conducted based on different fingerprint attack models, and classification accuracy was obtained by using the original data set, the defense method data set in this paper and the defense method based on differential evolution [18]. The experimental results are shown in Figure 5.

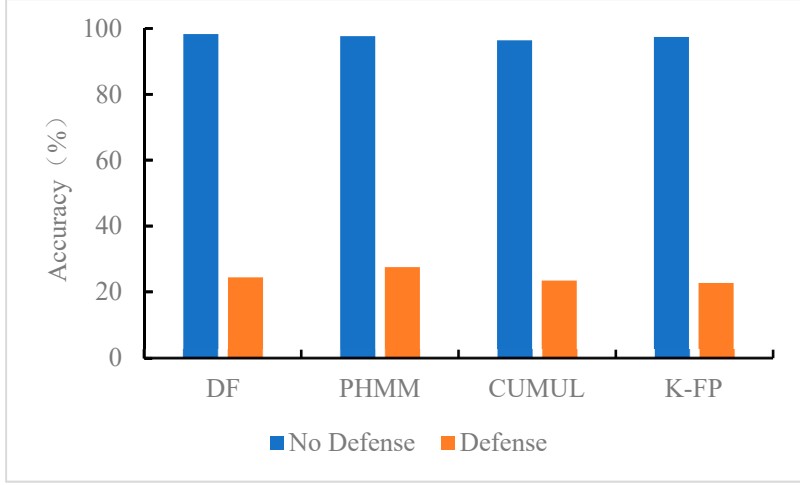

**Figure 5.** Experimental effects of various attack models.

It can be analyzed from Figure 3 that all kinds of attack modes can show a good classification effect on the original undefended data set, and the accuracy of the DF model and PHMM model is as high as 98.3% and 97.65%, respectively, which indicates that the fingerprint sequence modeling based on traffic can accurately capture the behavioral

characteristics of anonymous networks, and, therefore, poses a threat to anonymous communication. The accuracy of attack models is also above 90 percent. The classification accuracy of the defense method of the DE algorithm against DF and PHMM models is still about 50%. After the defense strategy in this paper is used, the accuracy decreases by more than 60%. This shows that the system can effectively defend against multiple traffic fingerprint attack models.

In addition, for further analysis, the classification effect of the verified fingerprint attack model under different iterations was measured. On the countermeasure sample data set, the DF model and the PHMM model, with better attack effects as mentioned above, were used, and ROC curves of the two classification models under different iterations were drawn in Figure 6.

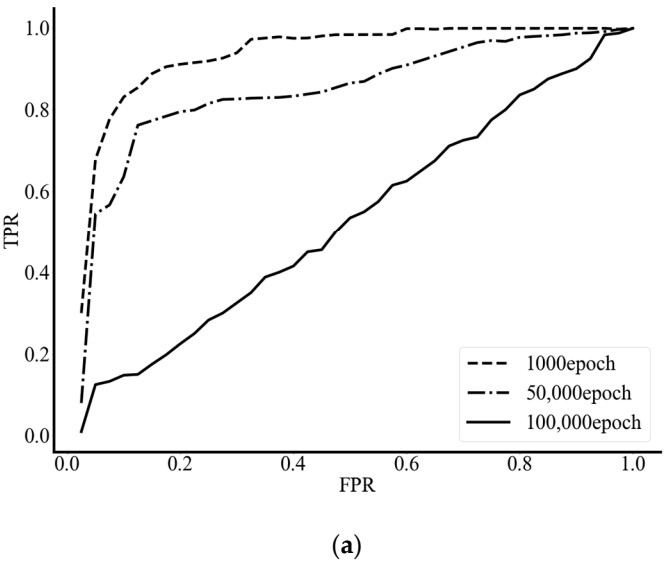

(**a**)

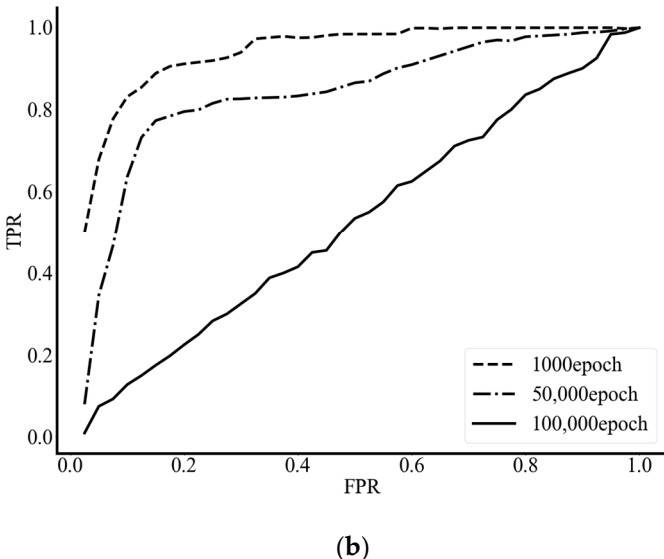

(**b**)

**Figure 6.** Model ROC curves of (**a**) PHMM model and (**b**) DF model.

It can be concluded from Figure 4 that for the two attack learning models, the defense effect is better when the number of iterations is increased. When the number of iterations is 1000, the website fingerprint attack model can distinguish the original traffic from the counter traffic. When the number of iterations reaches 100,000, the AUC values are 0.597 and 0.578. If the reviewers need to ensure that the true positive rate (TPR) is greater than

90%, the camouflaging traffic also has a probability of passing a review that requires close to 90%, indicating the actual effectiveness of the method.

### 5.3.3. Information Leakage Analysis

For the defense strategy experiment based on website fingerprinting attacks, we first comprehensively evaluate the performance of the defense strategy in terms of the misjudgment rate when facing the attack classifier model, and the load consumption when filling disturbance. We then use Shuai Li's website fingerprint information leakage measurement system, WeFDE, to analyze the data information leakage generated by the defense strategy used in this paper. We perform information leakage analysis on undefended full-duplex traffic and defended traffic for 3043 manually defined features spread across 14 categories. The leakage for each feature and defense is shown in Figure 7.

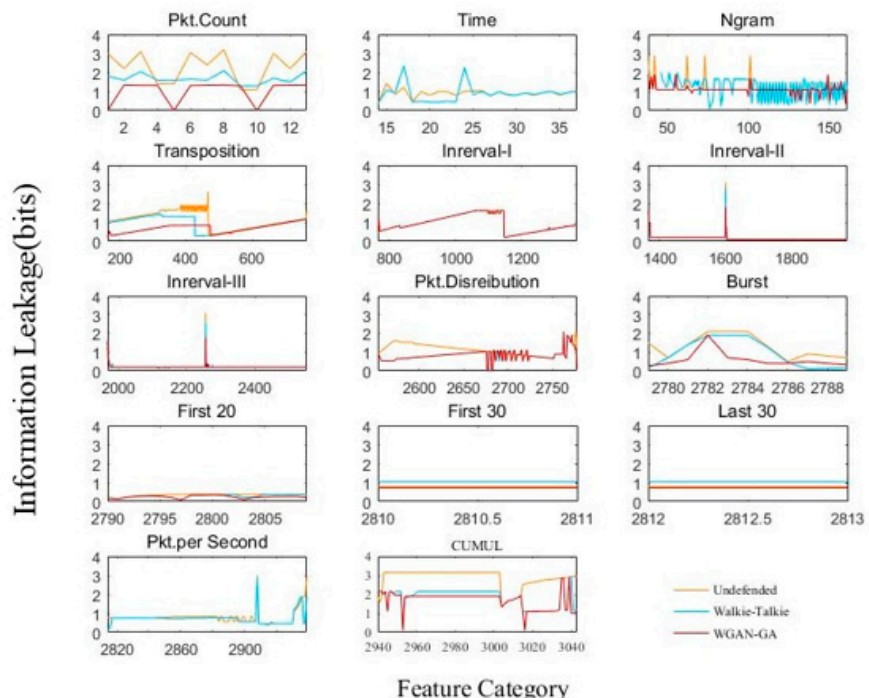

**Figure 7.** Individual feature information leakage.

Our Timing and Pkt. per Second categories do not include W-T or WGAN-GA measurements, as the simulations for these defenses are unable to produce accurate timestamp estimations.

In general, we find the information leakage of WGAN-GA to be comparable to the other defenses. We find that any given feature leaks at most 1.9 bits of information with WGAN-GA. Walkie-Talkie has a similar maximum leakage, with at most 2.2 bits per feature. The maximum amount of leakage seen for the undefended traffic was 3.4 bits, nearly twice that of WGAN-GA. Additionally, WGAN-GA shows notable improvements over Walkie-Talkie in the Transposition, First 30, Last 30, and Burst features, while Walkie-Talkie is better than WGAN-GA in the Pkt. Distribution feature. However, Li's [3] research has shown that packet length is the most leaked feature in the web fingerprint model, which further demonstrates the effectiveness of this method in defending against web fingerprint technology.

Overall, the results of our information leakage analysis are largely consistent with what we see in our accuracy measurements.

### 5.3.4. Comparison of Traffic Load Consumption

For a website fingerprint with a length of 3000, the proposed method performs unit filling at 30 locations, which indicates that the additional bandwidth consumption of the proposed method is 1%. The consumption pairs of defense models of the same type are shown in Table 3:

**Table 3.** Comparison of consumption pairs of defense models of the same type.

| Defense Models | WGAN-GA | DE | Walkie-Talkie |
|---|---|---|---|
| Bandwidth consumption | 1% | 0.8% | 31% |
| Time of convergence | 45.6 s | 113.6 s | 104.4 s |

The table shows that the WGAN-GA and DE algorithms screen the added disturbance when generating samples, which consumes very little bandwidth, while Walkie-Talkie uses the same strategy for each fingerprint, which will generate a large amount of redundant data, resulting in bandwidth loss. In terms of model training time, the proposed method is obviously superior to the DE algorithm.

### 6. Conclusions

We propose WGAN-GA, a WF defense that offers better protection and lower bandwidth overhead than DE and Walkie-Talkie, the previous state-of-the-art lightweight defenses. We reduce the accuracy of a variety of relatively advanced attack models from more than 90% to 20–30%. Such low accuracy is highly likely to cause the attacker to produce a large number of false positives, and our method is also more advantageous in information leakage analysis. We emphasize that our experiments are conducted in the closed-world setting, where the attacker knows that the Tor client is assumed to visit one of the monitored sites.

It is clear that Tor developers like adaptive padding defense because it is relatively easy to integrate into Tor code and has no latency overhead. Now, there are other ways [23] to attack Tor besides fingerprint attacks, and our experiment was conducted in a closed world. Thus, it is clear that there is much work to be performed before Tor users can feel confidently safe from the threat of traffic fingerprinting.

**Author Contributions:** Conceptualization, methodology and software, H.B. and J.Y.; validation, H.B., J.Y. and R.C.; formal analysis, H.B. and J.Y.; investigation, H.B. and J.Y.; resources, H.B. and J.Y.; data curation, H.B. and J.Y.; writing—original draft preparation, H.B.; writing—review and editing, H.B., R.C. and J.Y.; project administration, H.B. and R.C.; funding acquisition, J.Y. All authors have read and agreed to the published version of the manuscript.

**Funding:** This research received no external funding.

**Conflicts of Interest:** The authors declare no conflict of interest. This research does not involve any human or animal participation. All authors have checked and agreed with the submission.

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
