# Peer review of "Web Site Fingerprint Attack Generation Technology Combined with Genetic Algorithm"

_electronics, doi:10.3390/electronics12061449_

Round 1

Reviewer 1 Report

This manuscript talks about an anti-sample generation method using genetic algorithm to defend against fingerprint attacks on anonymous networks. 

The authors combined Wasserstein distance based adversarial generating network (WGAN) with genetic algorithm to screen the disturbance. Some related works regarding fingerprint attacks are described and short description about genetic algorithm (GA) is also mentioned. However, the authors should consider the following important points:

1. Detail explanation why it needs GA ?, why WGAN is combined with GA.

2. Why it involves anonymous network. What anonymous network means in this case. What's happening in the anonymous network. 

Usually, TOR and anonymous network have provided secure network with involvement of some cryptographic algorithms, why it needs anti-sample generation against fingerprint attacks. If the network involves the Internet, it may need security systems against any type of attacks in worldwide web including fingerprinting.

3. Regarding worldwide web application, there is a reference which always used, namely Open Worldwide Application Security Project (OWASP), https://owasp.org/. This is an online community that provides freely-available articles, methodologies, documentation, tools, technologies, etc in the field of web application security. And even, it updates categories about attacking type trend time by time.

Hence, many people considered to this security project related to the worldwide web applications for developing, investigation, etc.

However, the authors do not consider to this security project. So, what security references used in this manuscript must be explained in detail as well.

However, the authors target is defending against fingerprint attacks. But, the proof of security impact of the proposed scheme should be explained in detail rather than the explanation about GA. 

The target of the security system should also be clear, perhaps for transmission security, information security, or/and any other security target, and so on. 

Sometimes, the proofs with mathematical models can also be carried out. 

4. How strong the security and how big security impact of the proposed scheme. So, experiment about attacking scenarios should be conducted to prove it and compare to other existing similar schemes.

This will be even more convincing if proven by real implementations and its security investigations.

Reviewer 2 Report

The paper seems to show a novelty work but it is hard to follow sometimes. I recomend to check the redaction flow, and to improve the explanation and use of communication tools such as figures, equations, etc.

Authors must check carefully the folliwing aspects:

Regarding the references

Some references are never cited. Furthermore, they can be perceived as dubious sources. (e.g., ref. 20).

Ref. 23 is used for comparison between other methods in the experimental results. It is recommended to use a scientific paper as reference for this method instead of a presentation.

Some references are cited in the middle of their authors names (e.g., the first paragraph of section 4.1, ref. 23).

There are some characters used for the latex coding that are visible in the .pdf content for some references (e.g., ref. 22)

Acronyms

Some acronyms are never introduced (e.g. CUMUL), others are introduced after been used the first time (e.g., SVM), others are defined twice in the same paragraph (e.g., DE).

Language and Redaction

Some ideas are difficult to follow, compromiosing the communication of the content to the readers (e.g., last paragraph of the introduction). 

Figures and Tables

Figures should be explained in more details from the text.

Starting from Figure 2, there is a lack of coherence in the figures cited from the text and the figures number. For example, in the text is cited the Figure 3, refering to Figure 2, and so on.

Equations

The explanation of the link between the equations and their role in the research could be more developed in the text.

Reviewer 3 Report

According to the following, this manuscript is not scientifically mature. So, this manuscript in the present form should be submitted to the related conferences.

1- There is not much new scientific information.

2- There is no comparison with new journal papers.

3- Traffic classification attack models are not included in the manuscript.

4- 50% of references are related to conferences and the rest are not new.

Round 2

Reviewer 1 Report

The authors should address the previous comments as follows and more explanations needed.

1. Why it involves anonymous network. What anonymous network means in this case. What's happening in the anonymous network. 

Usually, TOR and anonymous network have provided secure network with involvement of some cryptographic algorithms, why it needs anti-sample generation against fingerprint attacks. If the network involves the Internet, it may need security systems against any type of attacks in worldwide web including fingerprinting.

Why GA algorithm is the solution, how GA algorithm has strong robustness ? How the mechanism of the algorithm to be integrated into TOR and anonymous network.

2. Regarding worldwide web application, there is a reference which always used, namely Open Worldwide Application Security Project (OWASP), https://owasp.org/. This is an online community that provides freely-available articles, methodologies, documentation, tools, technologies, etc in the field of web application security. And even, it updates categories about attacking type trend time by time.

Hence, many people considered to this security project related to the worldwide web applications for developing, investigation, etc.

However, the authors do not consider to this security project. So, what security references used in this manuscript must be explained in detail as well.

However, the authors target is defending against fingerprint attacks. But, the proof of security impact of the proposed scheme should be explained in detail rather than the explanation about GA. 

The target of the security system should also be clear, perhaps for transmission security, information security, or/and any other security target, and so on. 

Sometimes, the proofs with mathematical models can also be carried out. 

Why and how deep learning model can be related to the security system.

3. How strong the security and how big security impact of the proposed scheme. So, experiment about attacking scenarios should be conducted to prove it and compare to other existing similar schemes.

This will be even more convincing if proven by real implementations and its security investigations.

Reviewer 2 Report

Please, check the use of WF instead of WFP. If WF is another term, then should be defined. Otherwise, refer to the same concept using the same acronym when required.

Reviewer 3 Report

This manuscript can be accepted in the present form.

Round 3

Reviewer 1 Report

All responses to the previous comments regarding security issues and its impact should be inserted into the article in order to be recommended as an acceptable article. The editorial quality has been improved and the article's contribution can be accepted.
